# Network Pharmacology and Transcriptomics to Explore the Pharmacological Mechanisms of 20(S)-Protopanaxatriol in the Treatment of Depression

**DOI:** 10.3390/ijms25147574

**Published:** 2024-07-10

**Authors:** Xiangjuan Guo, Lili Su, Meiling Shi, Li Sun, Weijia Chen, Jianan Geng, Jianming Li, Ying Zong, Zhongmei He, Rui Du

**Affiliations:** College of Chinese Medicinal Materials, Jilin Agricultural University, Changchun 130118, China; gxj20002024@163.com (X.G.); 13630524570@163.com (L.S.); anxia143341@163.com (M.S.); 20221595@mails.jlau.edu.cn (L.S.); chenweijia_jlau@163.com (W.C.); gengjianan@jlau.edu.cn (J.G.); m15568781138@163.com (J.L.); zongying@jlau.edu.cn (Y.Z.)

**Keywords:** 20(S)-protopanaxatriol, depression, network pharmacology analysis, transcriptomics

## Abstract

Depression is one of the most common psychological disorders nowadays. Studies have shown that 20(S)-protopanaxatriol (PPT) can effectively improve depressive symptoms in mice. However, its mechanism needs to be further explored. In this study, we used an integrated approach combining network pharmacology and transcriptomics to explore the potential mechanisms of PPT for depression. First, the potential targets and pathways of PPT treatment of depression were screened through network pharmacology. Secondly, the BMKCloud platform was used to obtain brain tissue transcription data of chronic unpredictable mild stress (CUMS) model mice and screen PPT-altered differential expression genes (DEGs). Gene ontology (GO) analysis and the Kyoto Encyclopedia of Genes and Genomes (KEGG) analysis were performed using network pharmacology and transcriptomics. Finally, the above results were verified by molecular docking, Western blotting, and quantitative real-time polymerase chain reaction (qRT-PCR). In this study, we demonstrated that PPT improved depression-like behavior and brain histopathological changes in CUMS mice, downregulated nitric oxide (NO) and interleukin-6 (IL-6) levels, and elevated serum levels of 5-hydroxytryptamine (5-HT) and brain-derived neurotrophic factor (BDNF) after PPT treatment compared to the CUMS group. Eighty-seven potential targets and 350 DEGs were identified by network pharmacology and transcriptomics. Comprehensive analysis showed that transthyretin (TTR), klotho (KL), FOS, and the phosphatidylinositol 3-kinase–protein kinase B (PI3K-AKT) signaling pathway were closely associated with the therapeutic effects of PPT. Molecular docking results showed that PPT had a high affinity for PI3K, AKT, TTR, KL, and FOS targets. Gene and protein level results showed that PPT could increase the expression of PI3K, phosphorylation of PI3K (p-PI3K), AKT, phosphorylation of AKT (p-AKT), TTR, and KL and inhibit the expression level of FOS in the brain tissue of depressed mice. Our data suggest that PPT may achieve the treatment of depression by inhibiting the expression of FOS, enhancing the expression of TTR and KL, and modulating the PI3K-AKT signaling pathway.

## 1. Introduction

Depression is a common clinical mental disorder that is characterized by continuous and prolonged low mood, with high prevalence, high suicide rate, and high relapse rate [1]. According to statistics provided by the World Health Organization (WHO), there are currently approximately 280 million people suffering from depression globally, accounting for 3.8% of the world’s population [2], who increasingly tend to be younger and younger. It is expected that by 2030 the medical burden of depression will exceed that of tumors and cardiovascular diseases to become the world’s largest [3]. The pathogenesis of depression has not yet been clarified, and reliable biomarkers are lacking. Traditional antidepressants play an essential role in the treatment of depression. Still, they have a single mechanism of action, are prone to drug resistance, and have problems such as significant side effects, delayed efficacy, and ease of relapse after discontinuation of medication [4]. Therefore, the search for multi-targeted, safe, efficacious antidepressant drugs with few side effects has become a current problem to be solved.

Modern medicine believes ginseng has apparent effects on the nervous [5], cardiovascular [6], digestive [7], reproductive [8], and respiratory systems [9]. Ginsenosides, one of the main active components of ginseng, are divided into protopanaxadiol saponins, protopanaxatriol saponins, and oleanane saponins. 20(S)-protopanaxadiol (PPD) and 20(S)-Protopanaxatriol (PPT) are metabolites of propanaxadiol and propanaxatriol saponins in the human gut, which have more potent activity than their original saponins, so they have a wide range of applications. A recent study found that PPD can activate brain-type creatine kinase (CK-BB) and improve depression-like behavioral and structural plasticity impairment induced by corticosterone (CORT) injections [10]. Zhang et al. found that PPT can inhibit D-galactose (D-gal)-induced brain aging in mice by promoting mitochondrial autophagy flow [11]. Both PPD and PPT have protective effects on the central nervous system. PPT and PPD are similar in structure. At present, there are extensive studies on PPD in the treatment of brain injury. However, the mechanism of PPT in the treatment of depression combined with network pharmacology and transcriptomics has not been well studied.

Network pharmacology, transcriptomics, proteomics, metabolomics, and other technologies have been used to understand the multi-targeted therapeutic effects of drugs on disease models [12]. Network pharmacology takes a holistic view of biological networks and analyzes the connections between drugs, targets, and diseases [13]. The research concept of interpreting the law of molecular association between drugs and therapeutic objects coincides with the holistic idea of Chinese medicine. Transcriptomics, a technique used to study the transcriptome of an organism, demonstrates gene function and structure at a holistic level, reveals the molecular mechanisms underlying the role of drugs in disease, and has been used to identify differentially expressed genes (DEGs) in individuals [14]. Therefore, combining two modern biological analysis methods, network pharmacology and transcriptomics, can help elucidate herbal medicines’ active ingredients and molecular mechanisms to treat diseases [15].

This study used pharmacological techniques and molecular biology studies in chronic unpredictable mild stress (CUMS) mice to investigate PPT treatment effects [16]. In addition, network pharmacology and transcriptome analysis were used to identify potential targets of PPT for depression. These potential targets were validated using molecular docking, Western blotting, and quantitative real-time polymerase chain reaction (qRT-PCR). The results of this study are expected to provide new insights and ideas on the molecular mechanisms of PPT for depression.

## 2. Result

### 2.1. Weight Detection

As shown in Figure 1A, mice in the control group gained weight steadily during the experiment. Compared with the control group, the weight of mice in the CUMS group was significantly lower (*p* < 0.001), suggesting that depression can lead to weight loss in mice. Compared to the CUMS group, PPT treatment had a significant inhibitory effect on the weight loss of mice after two weeks (*p* < 0.001), indicating that PPT administration had a positive regulatory effect on mice.

### 2.2. Sucrose Preference Test (SPT)

After 28 days of modeling, the rate of sugar-water preference was decreased in the remaining groups of mice compared with the control group (*p* < 0.05). After the intervention, compared with the control group, the sugar-water preference rate of mice in the CUMS group was decreased (*p* < 0.05). Compared with the CUMS group, the sugar-water preference rate of mice in the PPT group was increased (*p* < 0.05), and the results are shown in Figure 1B.

### 2.3. Serum Biochemical Indicator Tests

5-HT, BDNF, IL-6, and NO are all closely related to depression and are vital indicators for judging the severity of depression. The results showed that the levels of IL-6 and NO in the CUMS group were elevated (*p* < 0.05) (Figure 1C,D), and the levels of BDNF and 5-HT were significantly reduced (*p* < 0.01) (Figure 1E,F) compared with the control group, which could be substantially improved by PPT treatment.

### 2.4. Morris Water Maze (MWM)

The water maze localization navigation trajectory diagrams are shown in Figure 1G. The escape latency of mice in the CUMS group was extremely significantly prolonged compared with that of the control group (*p* < 0.001), suggesting that there was an impairment in the spatial learning ability of these mice, which indicates that the modeling was successful. In contrast, there was a significant difference between the PPT group and the CUMS group (*p* < 0.05), suggesting that the PPT treatment had an improvement in the learning ability of CUMS mice (Figure 1H). In addition, the mice in the CUMS group had a highly significant decrease in the number of traversals across the original platform (*p* < 0.001) (Figure 1I) and a considerable reduction in the time spent in the original platform quadrant (*p* < 0.05) (Figure 1J), suggesting that the spatial memory ability of these mice was impaired. PPT treatment was able to increase the number of traversals and the time spent in the original platform quadrant of the CUMS mice significantly, suggesting that PPT could improve the spatial memory of CUMS mice impairment (*p* < 0.01, *p* < 0.05).

### 2.5. PPT Can Improve the Brain Histopathological Injury Induced by CUMS

The dentate gyrus (DG) area is the place in the brain where new neurons are generated, and it is a crucial area for regulating social behavior. The nerve cells in the hippocampal DG area of mice in the control group were tightly and orderly arranged, with clear outlines and normal structural morphology. Compared with the control group, some of the cells in the hippocampal DG area of mice in the CUMS group were disorganized and reduced in density, with apparent residual vacuoles of missing neurons, indicating a certain degree of pathological damage. Compared with the CUMS group, the hippocampal DG area of mice in the PPT group showed reduced pathology, more neurons, and relatively neat arrangement and dense layers. It indicated that PPT improved the pathological damage in the hippocampal DG area to a certain extent, and the results are shown in Figure 2A.

The CA3 region of the hippocampus is one of the most abundant regions in the hippocampus, which is essential for memory and navigation. HE staining results showed (Figure 2A) that neuronal cells in the CA3 region of the hippocampal tissue of the mice in the CUMS group were darkly stained with loose and deformed structures, reduced in number, and irregularly arranged compared with the control group. The neurons in the CA3 area of mice given PPT intervention were more structurally intact, with an increased number of cells and relatively neat and regular arrangement.

The number of nissl bodies is closely related to depression. Nissl bodies within the nerve cell cytosol decrease significantly after stimulation, and damage to neurons can be reflected by observing the number of nissl bodies. The results of nissl staining are shown in Figure 2B. The number of neurons and nissl bodies in the CA1 and CA3 regions of the hippocampal tissue of control mice was high, with rounded cell morphology, clear nuclei, and neat arrangement. Compared with the control group, the CUMS group showed a decrease in the number of nissl bodies (*p* < 0.05), cell deformation and crumpling, and blurred outlines, indicating more severe neuronal damage. After PPT administration, the number of nissl bodies in the CA1 and CA3 regions increased (*p* < 0.05), and the cell structure was intact, suggesting that PPT treatment could improve neuronal damage in CUMS mice.

### 2.6. Target Collection and PPI Network Analysis of PPT for Depression

Based on SwissTargetPrediction, PharmMapper, and ETCM databases, 145 PPT-related target sites were collected. Depression-related targets screened by GeneCards, OMIM, and DisGeNET databases were merged to remove duplicate values to obtain 3806 targets. A Venn diagram was produced by jvenn, and 87 potential targets of PPT for depression were obtained, as shown in Figure 3A. The PPI network consisted of chemical components, target proteins, and differential gene proteins, including 84 nodes and 493 edges, where nodes represent proteins and edges represent relationships between proteins (Figure 3B). According to the nature of the network topology, there were 17 key nodes with degree, betweenness centrality, and closeness centrality greater than the average (degree = 11.74, betweenness centrality = 100.93, closeness centrality = 0.006), accounting for 20.24% of the total number of nodes. The central core targets were ALB (degree = 45), PTGS2 (degree = 37), ESR1 (degree = 34), CYP3A4 (degree = 33), and FOS (degree = 29).

### 2.7. GO and KEGG Pathway Analysis

GO and KEGG pathway enrichment analyses were performed using the David platform to further explore the potential molecular mechanisms of PPT against depression. The GO enrichment analysis yielded 186 entries for biological processes (BP), 43 entries for cellular components (CC), and 76 entries for molecular functions (MF), totaling 305 entries. The top 10 significantly enriched items of BP, CC, and MF are visualized and analyzed in Figure 3C, and the entries related to biological processes mainly included intracellular steroid hormone receptor signaling pathway, positive regulation of gene expression, and signal transduction. The entries related to molecular function mainly included chromatin, endoplasmic reticulum membrane, and receptor complex. The entries regarding cellular composition mainly included RNA polymerase II transcription factor activity, ligand-activated sequence-specific DNA binding, steroid binding, and estrogen response element binding. KEGG enrichment analysis showed (Figure 3D) that 125 signaling pathways were mainly enriched in the chemical carcinogenesis-receptor activation pathways in cancer, the prolactin signaling pathway, the PI3K-AKT signaling pathway, and endocrine resistance. It was shown that PPT had a significant effect on these signaling pathways. In addition to this, a “target-pathway” network was constructed to further investigate the molecular mechanism of PPT against depression, as shown in Figure 3E. The network consists of 84 nodes and 235 edges. VDR, EPHX2, NR1I3, PIK3CD, and PIK3CB were highly gene-targeted. The results indicated that among the top 20 pathways of PPT for depression, chemical carcinogenesis-receptor activation pathways in cancer, the prolactin signaling pathway, the PI3K-AKT signaling pathway, and endocrine resistance should be the focus of further research.

### 2.8. Transcriptomics Analysis

To further explore the targets of PPT for depression, we performed transcriptome sequencing using the BMKCloud platform. Brain tissues from the control, CUMS, and PPT groups were selected for sequencing. Figure 4A shows that the CUMS group up-regulated 39 DEGs compared to the control group, and PPT could inhibit high expression of 8 DEGs. Meanwhile, compared to the control group, the CUMS group downregulated the expression of 151 DEGs, and PPT could activate low expression of 29 DEGs. PPT regulated a total of 37 DEGs. Hierarchical clustering maps of DEGs (Figure 4B) and heat maps of volcano maps (Figure 4C,D) were then constructed. In addition, DEGs were analyzed using GO and KEGG. As shown in Figure 5A, 389 PPT-regulated DEGs could be enriched in 128 GO entries when *p* < 0.05. There were 66 biological processes (BP), mainly including positive regulation of transcription from RNA polymerase II promoter, extracellular matrix organization, and cell–cell adhesion; 32 molecular functions (MF), mainly including extracellular space, extracellular exosome, and extracellular regions; 30 cellular localizations (CC), mainly including extracellular matrix structural constituents, transforming growth factor beta binding, and calcium ion binding. KEGG results showed (Figure 5B) that PPT affected the MAPK signaling pathway, protein digestion and absorption, the PI3K-AKT signaling pathway, vascular smooth muscle contraction, and proteoglycans in cancer.

### 2.9. Integrated Analysis of Transcriptome Sequencing and Network Pharmacology

Intersecting 87 targets predicted by network pharmacology for PPT for depression with 350 DEGs from transcriptome sequencing yielded a total of three target genes (TTR, FOS, and KL) (Figure 5C). Figure 5D shows that intersecting 125 pathways enriched by network pharmacology with 20 pathways enriched by transcriptome sequencing yielded a total of 8 pathways, i.e., the PI3K-AKT signaling pathway, proteoglycans in cancer, parathyroid hormone synthesis, secretion and action, fluid shear stress and atherosclerosis, diabetic cardiomyopathy, lipid and atherosclerosis, platelet activation, and pathogenic Escherichia coli infection.

### 2.10. Molecular Docking

Based on the results of the combined network pharmacology and transcriptome analysis, PI3K, AKT, TTR, FOS, and KL were selected as the receptors with 20(S)-protopanaxatriol for binding ability prediction, and the visualization results are shown in Figure 6. The magnitude of the binding energy can indicate the presence or absence of binding capacity between the ligand and the receptor. The larger the absolute value of the binding energy, the greater the stability. The data in Table 1 show that the binding energies of 20(S)-protopanaxatriol with each target are less than −6.0 kcal/mol, indicating a robust binding capacity. The above molecular docking results validated the accuracy of the network pharmacology and transcriptome screening.

### 2.11. Effects of PPT on PI3K-AKT Signaling Pathway and TTR, KL, and FOS Protein Expression in Depressed Mice

Network pharmacology and transcriptome sequencing analyses indicated that PPT exerts antidepressant effects that may correlate highly with the PI3K-AKT signaling pathway and TTR, KL, and FOS proteins. Therefore, the protein expression levels of PI3K, p-PI3K, AKT, p-AKT, TTR, KL, and FOS were determined by Western blotting (Figure 7A). The results of Western blotting showed that, compared with the control group, the ratios of p-PI3K/PI3K and p-AKT/AKT in the brain tissues of mice in the CUMS group were significantly lower (*p* < 0.001, *p* < 0.01) (Figure 7B,C), the relative expression of TTR and KL proteins were lower (*p* < 0.001) (Figure 7D,E), and the expression of FOS was significantly elevated (*p* < 0.01) (Figure 7F). Compared with the CUMS group, the brain tissues of mice in the PPT group showed a significant decrease in the p-PI3K/PI3K and p-AKT/AKT ratio, TTR and KL protein expression levels were elevated to different degrees (*p* < 0.05, *p* < 0.01) (Figure 7B–E), and PPT inhibited the protein expression level of FOS (*p* < 0.05) (Figure 7F).

### 2.12. qRT-PCR to Verify the Expression Level of Target Genes

To further validate the gene regulatory effects of PPT on the brain tissues of depressed mice, network pharmacology and transcriptome intersection differential genes (FOS, TTR, KL) and PI3K and AKT was selected. The experimental results, as shown in Figure 7G–K, showed that the PI3K mRNA expression level was significantly downregulated after CUMS stimulation (*p* < 0.01) (Figure 7G), the AKT mRNA expression level was decreased (*p* < 0.05) (Figure 7H), the TTR and KL mRNA expression levels were both significantly downregulated (*p* < 0.01) (Figure 7I,J), and the FOS mRNA expression level was upregulated (*p* < 0.05) (Figure 7K) compared with the control group. After PPT treatment, the above mRNA expression levels were reversed, and the resultant trend was consistent with the results of the Western blotting assay. This suggests that the mechanism of PPT treatment for depression may be related to the above targets and signaling pathways.

## 3. Materials and Methods

### 3.1. Reagents and Materials

20(S)-protopanaxatriol was purchased from Sichuan Weikeqi Biotechnology Co. (Sichuan, China). Saline was purchased from Jining Chenxin Pharmaceutical Co. (Jining, China). Interleukin-6 (IL-6) and brain-derived neurotrophic factor (BDNF) assay kits were purchased from the Shanghai Yuanju Biotechnology Center (Shanghai, China). A nitric oxide (NO) assay kit was purchased from Shanghai Biyuntian Biotechnology Co. (Shanghai, China). A 5-hydroxytryptamine (5-HT) assay kit was purchased from Hangzhou Pantechin Biotechnology Co. (Hangzhou, China). Phosphatidylinositol 3-kinase (PI3K), phosphorylation of PI3K (p-PI3K), protein kinase B (AKT), phosphorylation of AKT (p-AKT), transthyretin (TTR), and FOS antibodies were purchased from Shenyang Wanlei Biotechnology Co. (Shenyang, China). Klotho (KL) antibody and horseradish peroxidase (HRP)-labeled goat anti-rabbit lgG (H + L) were purchased from Beijing Bioss Biotechnology Co. (Beijing, China). The bicinchoninic acid (BCA) protein detection kit was purchased from Shanghai Yuanye Biotechnology Co. (Shanghai, China). PCR primers were designed and synthesized by Wuhan Sevier Biotechnology Co. (Wuhan, China). Morris water maze analysis system was purchased from Chengdu Taimeng Software Co. (Chengdu, China).

### 3.2. Animal Models and Therapy

C57BL/6 male mice (18 ± 2 g) were purchased from Changchun Yise Laboratory Animal Technology Co. (Changchun, China). The research project was reviewed by the Laboratory Animal Welfare and Ethics Committee of Jilin Agricultural University and found to align with the ethical needs of laboratory animals. Ethical review acceptance number: 20211011003 (Jilin Agricultural University Laboratory Animal Center Laboratory Animal License: SYXK (ji) 2018–2023). All mice were acclimatized and fed for 3 days with normal light and humidity and free access to water and food. Thirty mice were randomly divided into three groups (*n* = 10): the control group, the CUMS group, and the PPT group. In addition to the control group, the other groups of mice were subjected to CUMS stimulation for 4 weeks: 24 h of fasting, 24 h of water fasting, 12 h of tilting the rat cage, 5 min of swimming in warm water, 5 min of swimming in icy water, 12 h of damp bedding, day and night reversal, 3 h of barking of natural enemies, 12 h of stimulation by foreign objects, 12 h of the empty cage, each animal received 2 stimuli per day, fasting and water fasting did not co-occur, and each stimulus was not repeated for 3 days. The extent of the effect of CUMS on mice was assessed using weight and behavioral tests.

After administration by intragastrical gavage from week 5 onwards, the control group and CUMS group mice were gavaged with 0.2 mL/d saline, and mice in the PPT group were given an equal volume of 40 mg/kg/d of PPT [17], lasting two weeks. On the last day of the experiment, blood was taken from the eyeballs. After decapitation and execution, brain tissues from all three groups of mice were collected and immediately stored in liquid nitrogen for rapid freezing and then placed in a refrigerator at −80 °C awaiting use.

### 3.3. Weight Detection

The experiment lasted for six weeks, and the weight of mice in each group was monitored weekly during the whole experiment.

### 3.4. Sucrose Preference Test (SPT)

The characteristic symptoms of depression are pleasure deficit, lack of interest, and depressed mood. SPT is the classic test for detecting symptoms of pleasure deficit in depression. All mice were given a water bottle containing 2% sugar water 24 h before the experiment to familiarize themselves with the taste of sugar water. Mice were deprived of water and food for 12 h before the formal experiment and then given a bottle of 2% sugar water and a bottle containing normal drinking water, switching the positions of the two bottles after 12 h to avoid interference from the position of the bottles. The consumption of sugar water and normal drinking water was measured by recording the change in bottle weight over a 24 h period [18]. The formula is as follows: sugar water preference rate (%) = sugar water consumption/(sugar water consumption + drinking water consumption) × 100%.

### 3.5. Serum Biochemical Indicator Tests

The serum was separated after blood sampling from mouse eyes and placed in a −80 °C refrigerator for storage. The serum levels of 5-HT, BDNF, IL-6, and NO were detected by the ELISA method according to the kit instructions.

### 3.6. Morris Water Maze (MWM)

The Morris water maze apparatus was divided into 4 quadrants, and a platform 1 cm below the horizontal was placed in the Ⅰ quadrant and kept stationary. The temperature of the water tank was set at 25 °C. It was divided into a localization navigation experiment and a spatial search experiment. A localization navigation experiment was used to test the spatial learning ability of mice, and a spatial search experiment was used to test the spatial memory ability of mice. The experiments were conducted for 4 days, with 3 training sessions per day, the first 3 days for the localization navigation experiment and the last day for the spatial search experiment.

#### 3.6.1. Localization Navigation Experiment

The mice were gently placed in the remaining three quadrants except for the quadrant where the platform was located in order, and they were observed and timed for 300 s. The time required for the mice to climb up to the platform after entering the water was recorded, i.e., the latency to escape. The mice were removed from the water and wiped dry at the end of the experiment when they had found the platform and stayed on it for 5 s, and then were placed in the rat cage. If the mice did not find the platform within 300 s, the experiment was automatically ended, and the mice were artificially guided from the water to the platform and stayed on it for 10 s to familiarize themselves with the surroundings of the Morris water maze system.

#### 3.6.2. Spatial Search Experiment

At the end of the localization navigation experiment, the platform was removed from the water. The mice were gently placed in the quadrant opposite the one where the platform was located, and the time they entered the quadrant in which the original platform was located and the number of times they crossed the original platform within 300 s were recorded.

### 3.7. Histopathological Examination

Brain tissues from each mouse were fixed in 4% paraformaldehyde for 24 h, washed with water for 4 h, dehydrated with graded ethanol, embedded in paraffin according to standard histological procedures, sectioned at a thickness of 4 μm, and then stained with HE to observe the histomorphological changes and pathological changes under the microscope [19].

The damage of neurons in the hippocampus was observed by nissl staining. Paraffin sections of brain tissue were dewaxed to water, stained, dehydrated, transparent, sealed, and the staining results were observed under a microscope.

### 3.8. Target Collection and PPI Network Analysis of PPT for Depression

The English name, structure data file (SDF) structural formula, and simplified molecular input line entry system (SMILES) number of the PPT were obtained from the PubChem database (https://pubchem.ncbi.nlm.nih.gov/ (accessed on 24 November 2023)) [20]. The information of PPT was uploaded to SwissTargetPrediction (http://www.swisstargetprediction.ch/ (accessed on 24 November 2023)) [21], PharmMapper (https://pubmed.ncbi.nlm.nih.gov/28472422/ (accessed on 24 November 2023)) [22], and Encyclopedia of Traditional Chinese Medicine (ETCM) (http://www.tcmip.cn/ETCM/ (accessed on 24 November 2023)) [23] databases for target prediction analysis. The PPT duplicate genes obtained from the above three databases were integrated and deleted to obtain the relevant PPT targets. Using “depression” as a keyword, we screened the GeneCards (https://www.genecards.org/ (accessed on 25 November 2023)) [24], Online Mendelian Inheritance in Man (OMIM) (https://omim.org/ (accessed on 25 November 2023)) [25], and DisGeNET (https://www.disgenet.org/ (accessed on 25 November 2023)) [26] databases for relevant targets of depression. The targets were merged to remove duplicate genes and obtain target information for depression. Therapeutic targets for PPT against depression were obtained using the jvenn online tool (https://jvenn.toulouse.inrae.fr/app/example.html (accessed on 25 November 2023)) [27].

The predicted targets of PPT for depression were imported into the Search Tool for the Retrieval of Interacting Genes (STRING) database (https://www.string-db.org/ (accessed on 26 November 2023)) [28]. Protein–protein interaction (PPI) network relationships were constructed, and medium confidence > 0.4 was used to improve the accuracy of the PPI network. Protein interaction information was imported into Cytoscape software (version 3.9.1) for graphical visualization and analysis. The nodes’ size and color reflected the degree values, which were categorized and sorted according to the magnitude of the degree values. The importance of each node in the PPI network was assessed using degree, betweenness centrality, and closeness centrality.

### 3.9. GO and KEGG Pathway Analysis

The Database for Annotation, Visualization, and Integrated Discovery (DAVID) database integrates several database resources and is mainly used to identify differences in the function of genes and pathway enrichment analysis. The DAVID database ((https://david.ncifcrf.gov/ (accessed on 27 November 2023)) [29] was used for gene ontology (GO) enrichment and Kyoto Encyclopedia of Genes and Genomes (KEGG) pathway analysis of potential targets of PPT for depression [30], with the species restriction of “Homo sapiens” and a threshold value of *p* < 0.05, and the results were ranked according to the *p*-value. Potential targets were analyzed using three GO modules to annotate gene functions: biological process (BP), molecular function (MF), and cellular component (CC), each describing the biological process in which the gene product may be involved, the molecular function it exercises, and the cellular environment in which it resides. KEGG pathway enrichment analysis was used to predict the potential biological functions and molecular pathways regulated by PPI. The GO and KEGG pathway enrichment analysis results were visualized and analyzed in the bioinformatics platform ((https://www.bioinformatics.com.cn/ (accessed on 27 November 2023)) [31]. To explore the interaction relationship between target components and signaling pathways, Cytoscape software (version 3.9.1) was used to construct “target-pathway” network diagrams, with nodes denoting targets or signaling pathways and connecting lines indicating the interaction relationship between nodes.

### 3.10. RNA Sequencing (RNA-Seq) and Data Analysis

The hippocampal tissues of 3 mice in each group were collected for RNA-seq by the Beijing BMKCloud platform. Samples of total RNA were extracted from each sample using an RNA purification kit. Complementary DNA (cDNA) synthesis was performed after fragmentation of total RNA fragments into short fragments and enrichment of messenger RNA (mRNA) using oligo (dT) magnetic beads.

Differential expression analysis was performed using DESeq2_edgeR, and DEGs were obtained using fold change (FC) ≥ 1.5 and *p*-value ≤ 0.05 as the screening criteria. Volcano plots and cluster heatmaps of differentially expressed genes were drawn using the bioinformatics platform ((https://www.bioinformatics.com.cn/ (accessed on 1 December 2023)) [31]. GO enrichment analysis and KEGG enrichment analysis were performed on differentially expressed genes using the DAVID database ((https://david.ncifcrf.gov/ (accessed on 3 December 2023)) [29] to obtain relevant target functions and perform pathway analysis.

### 3.11. Molecular Docking

A molecular docking technique was utilized to verify the binding ability between PPT and critical targets. The 2D structure file of PPT was obtained from the PubChem database (https://pubchem.ncbi.nlm.nih.gov/ (accessed on 5 December 2023)) [20] and transformed into 3D confirmation by Chem3D 21.0.0 software to obtain the pdb file of small molecules. The protein structure of key targets was searched and downloaded from the PDB database (https://www.wwpdb.org/ (accessed on 5 December 2023)) [32], and the small molecule ligands and water molecules of each key target were deleted using Pymol 2.4.0 software. Finally, the processed PPT and key targets were molecularly docked using Auto Dock Vina 4.2 software to analyze the binding relationship between PPT and critical targets.

### 3.12. Western Blotting

Total proteins were extracted from brain tissues using a radioimmunoprecipitation assay (RIPA) lysis buffer, and the protein concentration of the samples was determined by the BCA method. The samples were mixed with a 5× sodium dodecyl sulfate-polyacrylamide gel electrophoresis (SDS-PAGE) uploading buffer in a ratio of 4:1, and boiling water was boiled for 10 min to denature the proteins. Approximately 10 µg of protein were placed on a 10% SDS-PAGE gel for electrophoretic separation. After electrophoresis, the separated target proteins were transferred to polyvinylidene fluoride (PVDF) by the wet transfer method and closed with protein-free rapid closure solution for 5 min at room temperature. Then, PI3K (1:1500), p-PI3K (1:800), AKT (1:800), p-AKT (1:800), TTR (1:800), KL (1:1000), FOS (1:500), and β-actin (1:800) primary antibody were added and incubated overnight at 4 °C. They were washed thrice with tris buffered saline with Tween 20 (TBST) for 10 min each time. HRP-labeled goat anti-rabbit lgG (H + L) (1:8000) was added and incubated at room temperature for 2 h on a shaker, and the photo was developed with an enhanced chemiluminescence (ECL) developer. The gray value of the bands was determined using ImageJ software v1.53q (National Institutes of Health, Bethesda, ML, USA), and the ratio of the target protein to the internal reference β-actin was used as its relative content.

### 3.13. qRT-PCR to Verify Gene Expression Levels

To verify the reliability of the target results, qRT-PCR was used to detect the expression level of target genes in tissues. GAPDH was used as the internal reference gene, and the relative expression of the gene was calculated using the 2^−ΔΔCT^ method. The sequences of the target genes were searched on the National Center for Biotechnology Information (NCBI) database ((https://www.ncbi.nlm.nih.gov/ (accessed on 3 November 2023)) [33], and the primer sequences are shown in Table 2.

### 3.14. Statistical Analysis

All data are expressed as mean ± SD. Differences between groups were analyzed using a one-way analysis of variance (ANOVA) followed by a *t*-test. Data were statistically analyzed using GraphPad Prism 9.5.1. *p* < 0.05 was considered statistically significant.

## 4. Discussion

Depression is a chronic relapsing mental illness that severely impairs patients’ quality of life [34]. At present, the etiology and pathogenesis of depression have not been fully clarified, and Western drug therapy occupies a significant position. Still, because Western drug therapy is characterized by addiction and a high relapse rate, there is an urgent need for new and effective antidepressant drugs [35]. Studies have shown that PPT vigorously protects the nervous system [36] and possesses multi-target and multi-pathway pharmacological properties. However, the underlying mechanisms remain unclear. Network pharmacology and transcriptome sequencing are common techniques for studying the molecular mechanisms of drugs [16]. To clarify the mechanism of action of PPT, network pharmacology and transcriptome sequencing technologies were used to predict potential molecular targets, and the results of the study were rationally analyzed, which can help to explore further the core targets and mechanism of action of PPT in the treatment of depression [37].

In this study, we investigated the efficacy and mechanism of PPT to improve depression by assessing changes in depression-like behaviors and depression-related indicators in a depression-like CUMS mouse model. The CUMS approach has been applied to the induction of depression in animal models, mimicking stressful events in daily human life [38]. In the present study, CUMS stimulation caused weight loss and induced significant depression-like behavior in mice. Mice in the CUMS group exhibited lower sucrose consumption in the sugar-water preference experiment compared to the control group. In the Morris water maze experiment, the mice in the CUMS group had significantly longer escape latencies. Their time in the original platform quadrant and across the original platform was reduced considerably. All these results indicated that the mice in the CUMS group had depressive symptoms. Depressed mice exhibiting despairing behavior could be significantly improved after PPT intervention, thus alleviating their depressive symptoms.

To date, there are a lack of effective treatments for depression because the etiology of the disorder is unclear, and the pathogenesis is very complex. Currently, the mechanism research takes neurotransmitters, inflammatory factors, neuroplasticity-related signaling pathways, and other mechanisms as the primary research direction. Previous studies have shown that decreased serum BDNF levels in patients may be predictive of major depression and that increased BDNF levels are associated with antidepressant effects [39]. Some studies have found that plasma NO concentrations in depressed patients are significantly correlated with the severity of depressive symptoms. NO is associated with developmental and neuroprotective effects. Still, high or sustained concentrations of NO not only directly damage nerve cells but also cause a series of inflammatory cascade responses and interfere with various molecular signaling pathways. Inflammatory responses are strongly associated with depression [40]. Previous analyses have noted that pro-inflammatory cytokines such as TNF-α and lL-6 are significantly increased in depressed patients [41]. Depression is also associated with reduced levels of monoamine neurotransmitters, such as 5-HT. Therefore, in addition to behavioral tests, to investigate the potential correlation between PPT and depression, we also examined the levels of BDNF, NO, IL-6, and 5-HT. In our results, BDNF, NO, IL-6, and 5-HT showed the same trend results as previously reported, which validates the model’s success and demonstrates the role of PPT in improving depression.

The hippocampus is a critical brain region for forming memories. The hippocampal DG area is a key region for memory retrieval and updating [42]; neuronal damage and impaired regeneration in the DG area may be essential pathological hallmarks in the development of depression. Neurons in the CA1 region of the hippocampus play a crucial role in memory formation, consolidation, and retrieval [43]. CA3 area plays a vital role in spatial working memory and associative memory and is one of the wealthiest areas in the hippocampus [44]. Nissl bodies are one of the characteristic structures of neurons, and their number can reflect the functional strength of neurons [45]. When neurons are damaged by stimulation, the number of nissl bodies decreases or even disappears. HE staining results showed that the number of neuronal cells in the DG and CA3 regions of the mouse hippocampus decreased, and the arrangement was sparse and disorganized, suggesting that the mouse hippocampal neurons had undergone certain pathological changes, the administration of PPT increased the number of neuronal cells, and the arrangement was more regular. The results of nissl staining showed that PPT increased the number of nissl bodies in CA1 and CA3 regions and improved the function of neurons. The above results indicated that PPT could reduce the pathological damage and alleviate the depression-like symptoms in CUMS mice.

To further explore the potential molecular mechanisms of PPT against depression, network pharmacology was initially used to predict potential targets and signaling pathways. KEGG results of network pharmacology showed that the PI3K-AKT signaling pathway, among others, was involved in the protective effect of PPT against depression. Transcriptomics is another bioinformatics approach that allows the study of gene expression at the RNA level. Therefore, we also used a transcriptomics approach to explore potential signaling pathways. In the transcriptome sequencing analysis, 126 upregulated genes and 263 downregulated genes were involved in the protective effect of PPT against depression. Notably, the KEGG pathway enrichment analysis of the transcriptome sequencing analysis provided the same results as the network pharmacology analysis that the PI3K-AKT signaling pathway is involved in regarding the effect of PPT on depression. A PI3K-AKT signaling pathway is a pro-survival signaling pathway that regulates neuronal cell growth, proliferation, differentiation, and apoptosis in the central nervous system. Activation of PI3K phosphorylates phosphatidylinositol on cell membranes, which in turn generates phosphatidylinositol triphosphate, leading to the full activation of AKT [46], and AKT phosphorylation activates neuroprotection [47]. Activation of the PI3K-AKT signaling pathway is protective of the central nervous system in several studies [48,49,50]. These reports are consistent with our results. Considering the consistent results of the transcriptomic analysis, the network pharmacology analysis, and the scientific evidence obtained to date [51], we hypothesized that the PI3K-AKT signaling pathway might be involved in the antidepressant effects of PPT.

By combining network pharmacology and transcriptomic analyses, we finalized the identification of three key targets (TTR, FOS, and KL). TTR is a neuroprotective protein that is important for brain growth and development. FRYE et al.’s [52] clinical study found that serum TTR concentrations were significantly lower in depressed patients than in controls. Overexpression of FOS leads to neuronal cell damage, cell death, and apoptosis [53]. KL has a crucial neuroprotective role in the hippocampus, and its overexpression enhances learning and memory [54]. In our study, the overall trends of TTR, FOS, and KL found by Western blotting and qRT-PCR coincided with those of the existing studies, suggesting that PPT can improve the central nervous system and exert a neuroprotective effect on neuronal cells.

In summary, the present study systematically elucidated the potential mechanism of PPT in treating depression by combining network pharmacology and transcriptomics techniques. It is hypothesized that PPT may delay the development of depression in mice by regulating the PI3K-AKT signaling pathway. This study provides a theoretical basis for the research and application of PPT in the field of antidepressants.

## 5. Conclusions

Based on the combination of network pharmacology with transcriptomic and experimental validation, this study elucidated the key targets and specific mechanisms of PPT for depression from multiple perspectives. The binding ability of the above targets to PPT was verified by the molecular docking technique. The above targets’ protein and gene expression levels were further verified using Western blotting and qRT-PCR. We concluded that PPT might be therapeutic in depression through the PI3K-AKT signaling pathway and unearthed three key targets (TTR, FOS, and KL). This study provides a novel and comprehensive strategy for the research and treatment of depression, and this strategy can also be used for the analysis of other diseases and drugs.

## Figures and Tables

**Figure 1 ijms-25-07574-f001:**
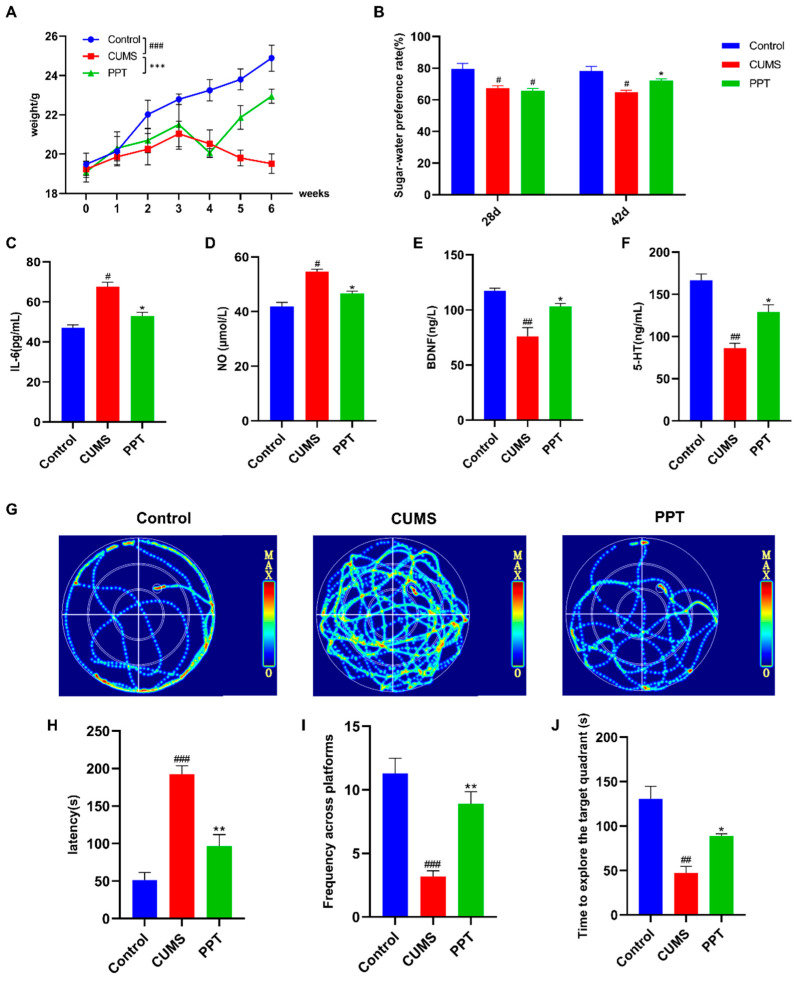
Effects of PPT on depression-like behavior and serum biochemical indices in mice. (**A**) Mouse weight change curve. (**B**) Sugar-water preference rate of mice. 28d: CUMS molding takes 28 days. 42d: Sugar water preference rate of mice after 42 days of CUMS modeling and 14 days of PPT intervention. (**C**–**F**) Expression levels of IL-6, NO, BDNF, and 5-HT in mouse serum were detected by ELISA. (**G**) Diagram of the experimental trajectory of the mouse water maze localization navigation experiment. (**H**) Escape the incubation period. (**I**) Number of trips across the original platform. (**J**) Dwell time in the original platform quadrant. Compared to the control group *p*
^#^ < 0.05, *p*
^##^ < 0.01, and *p*
^###^ < 0.001. Compared to the CUMS group: *p* * < 0.05, *p* ** < 0.01, and *p* *** < 0.001.

**Figure 2 ijms-25-07574-f002:**
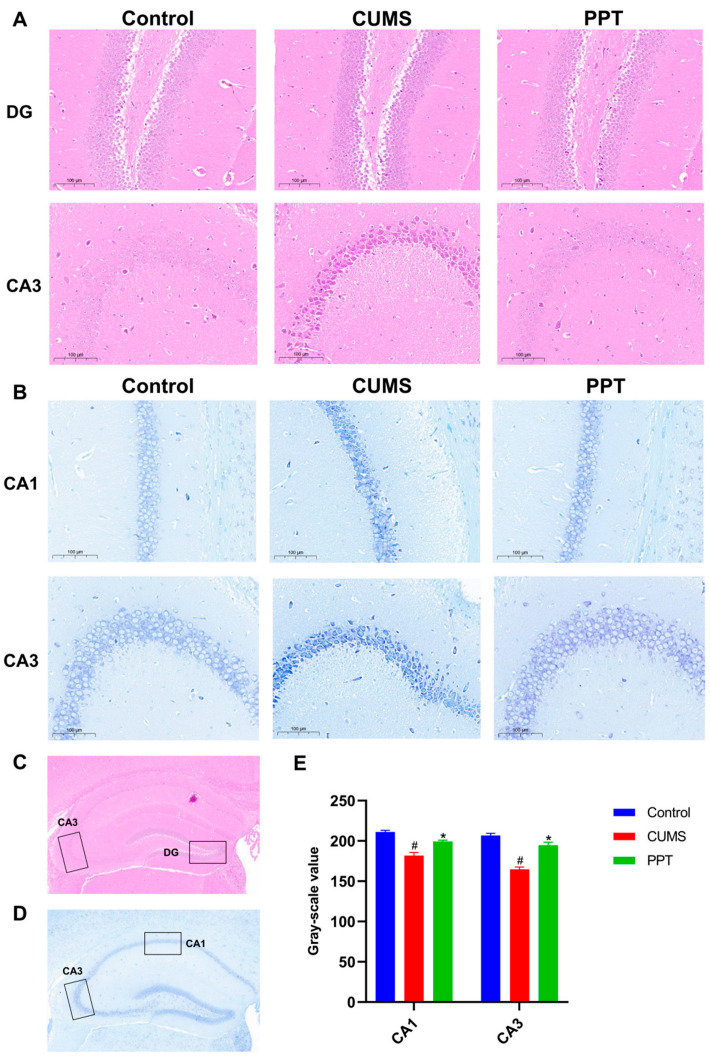
Effect of PPT therapy on brain histopathological changes induced by CUMS. (**A**) HE staining in each group’s DG region and CA3 region of hippocampal tissue, scale = 100 µm. (**B**) Representative images of CA1 and CA3 regions of hippocampal tissue in each group with nissl staining, scale = 100 µm. (**C**) Diagram of different regions stained by HE. (**D**) Schematic diagram of the different regions of nissl staining. (**E**) Quantification of gray-scale value of nissl bodies in CA1 and CA3 regions by nissl staining. Compared with the control group, *p*
^#^ < 0.05. Compared with CUMS group, *p* * < 0.05.

**Figure 3 ijms-25-07574-f003:**
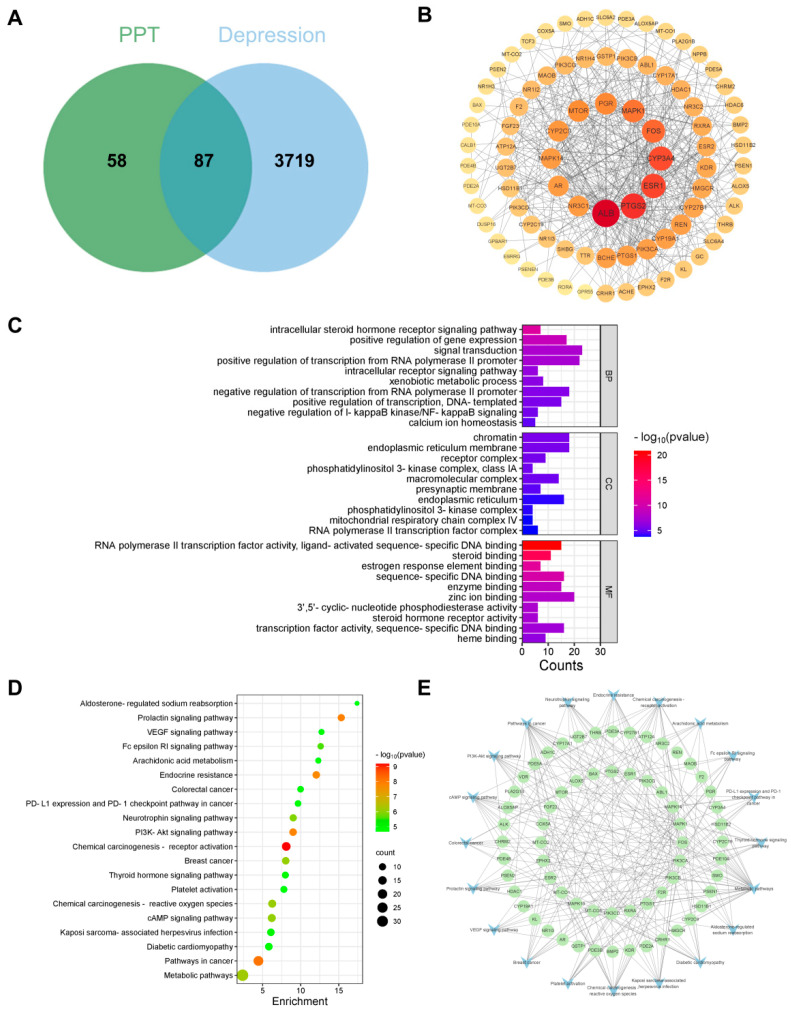
Results of a network pharmacology analysis. (**A**) PPT treatment of depression Venn diagram. (**B**) PPI network of potential targets for PPT treatment of depression, the connecting lines indicate their interactions, and the node sizes and color shades are proportional to their degree. (**C**) GO enrichment analysis of PPT for depression. (**D**) KEGG enrichment analysis of PPT for depression. (**E**) The “target-pathway” network diagram has green nodes representing potential protein targets and blue nodes representing pathways.

**Figure 4 ijms-25-07574-f004:**
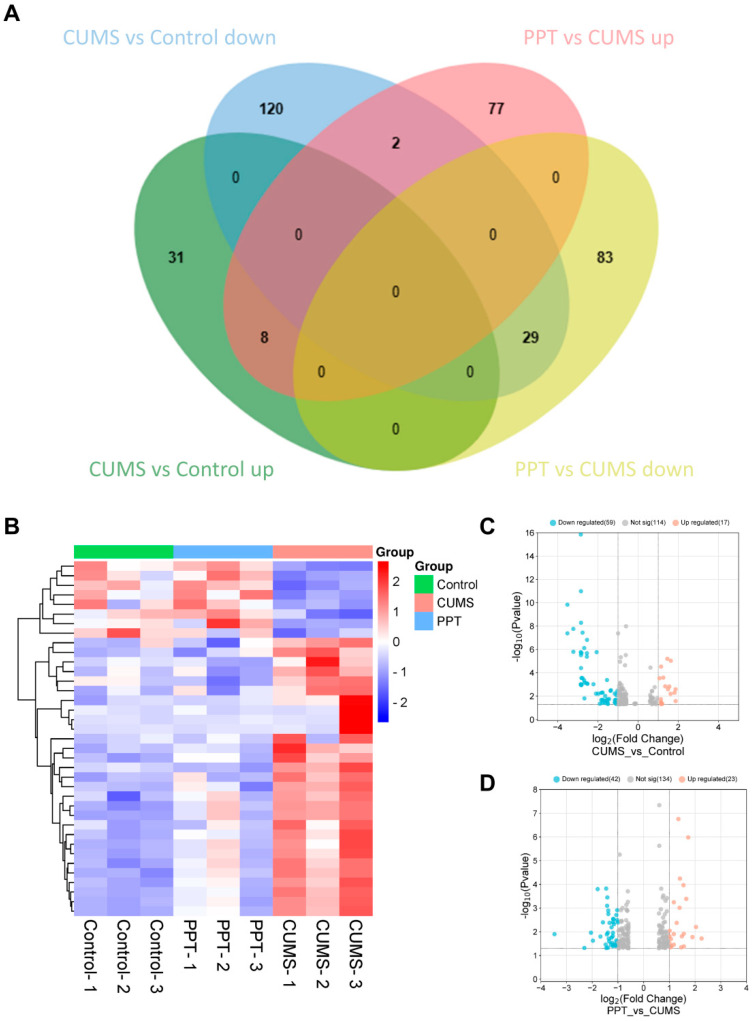
Transcriptomic analysis of DEGs. (**A**) DEGs Venn diagram. (**B**) Heatmap for cluster analysis of DEGs. (**C**,**D**) Volcano map of DEGs.

**Figure 5 ijms-25-07574-f005:**
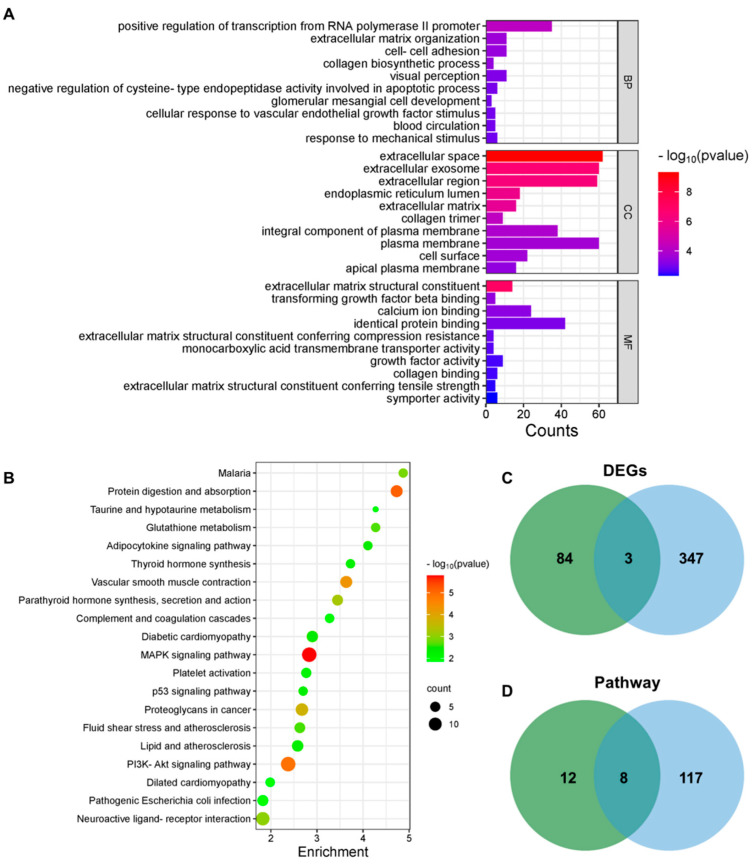
Transcriptomic analysis of GO and KEGG enrichment and combined network pharmacology and transcriptome analysis of Venn diagrams. (**A**) GO enrichment analysis of DEGs. (**B**) KEGG enrichment analysis of DEGs. (**C**) Network pharmacology and transcriptome intersection targets Venn diagram. (**D**) Network pharmacology and transcriptome intersection pathway Venn diagrams.

**Figure 6 ijms-25-07574-f006:**
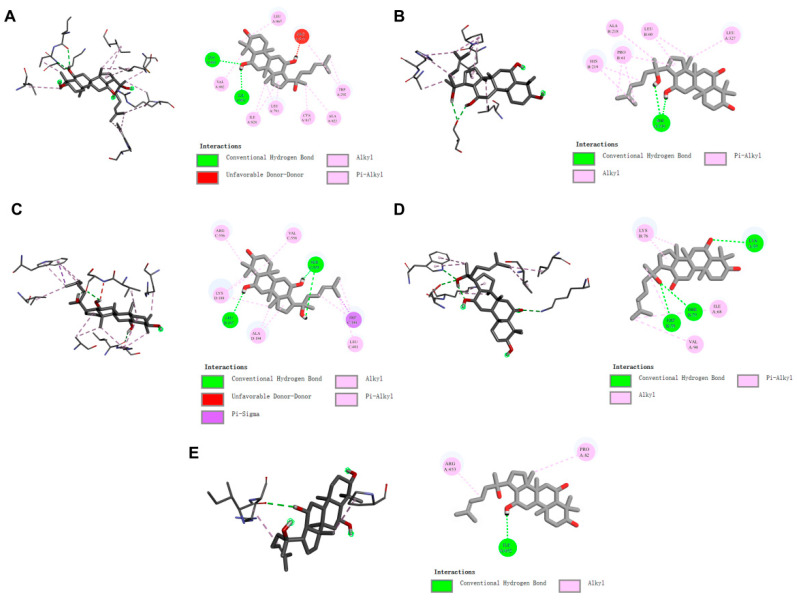
Molecular docking visualization results. (**A**) PI3K. (**B**) AKT. (**C**) FOS. (**D**) TTR. (**E**) KL. The stick structure is shown on the left, and the interaction is shown in 2D on the right.

**Figure 7 ijms-25-07574-f007:**
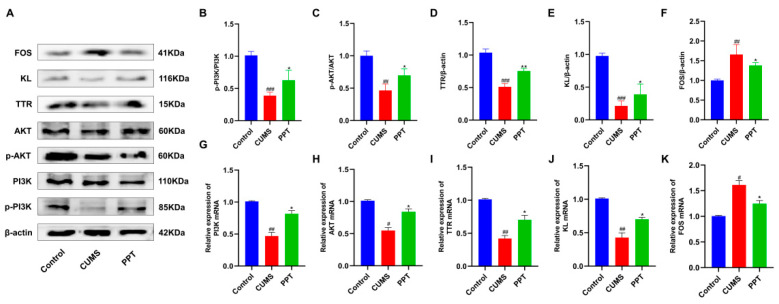
Preliminary validation of the mechanism of action of PPT for depression. (**A**) Western blotting to detect the expression levels of PI3K-AKT signaling pathway proteins and TTR, KL, and FOS proteins. (**B**–**F**) Histograms showing the protein expression levels of p-PI3K/PI3K and p-AKT/AKT ratio, TTR, KL, and FOS. (**G**–**K**) qRT-PCR detection of PI3K, AKT, TTR, KL, and FOS mRNA expression in mouse brain tissue. Compared with the control group, *p*
^#^ < 0.05, *p*
^##^ < 0.01, and *p*
^###^ <0.001. Compared with the CUMS group, *p* * < 0.05 and *p* ** < 0.05.

**Table 1 ijms-25-07574-t001:** Molecular docking score.

Compound	PDB ID	Molecular Docking Score (kcal/mol)
PI3K	1E8X	−7.9
AKT	7WM1	−7.6
FOS	1S9K	−8.8
TTR	3D7P	−7.6
KL	5VAK	−6.8

**Table 2 ijms-25-07574-t002:** Primer sequences.

Gene Name	Primer Sequences (5′ → 3′)
GAPDH	F:GCCTCCTCCAATTCAACCCT
	R:CTCGTGGTTCACACCCATCA
PI3K	F:ACCTTAAATGGTGAGCACGGA
	R:GGCCCGCACTGTAACCTATT
AKT	F:CCGCCTGATCAAGTTCTCCT
	R:AGAGGGAGAGGGCCAGTTAG
TTR	F:TTCCGTCTGCTCCTCCTTT
	R:AACACCTTCACACCCACATTC
FOS	F:CTGAGTGTCTCAAGTGCCTC
	R:CTCGGGTTGTAGGATTGAG
KL	F:CCTCCTTTACCTGAGAACCA
	R:GCACATCCCACAGATAGACA

## Data Availability

Data will be made available on request.

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
