# Peer review of "Network Pharmacology and Transcriptomics to Explore the Pharmacological Mechanisms of 20(S)-Protopanaxatriol in the Treatment of Depression"

_ijms, 2024, doi:10.3390/ijms25147574_

Round 1

Reviewer 1 Report

Comments and Suggestions for Authors

This work by Guo et. al, mainly investigates the effects of 20(S)-protopanaxatriol (PPT) on depression using network pharmacology and transcriptomics. In general, this is an interesting topic, but several major issues need to be resolved. 

1) There have been many similar and related studies in this direction. However, the authors failed to mention them in the introduction part. 

A very recent study J. Agric. Food Chem. 2024, 72, 10376 investigates the 20(S)-Protopanaxadiol [PPD, very similar to PPT] and its antidepressive effect. 

The authors also failed to mention PPD, which has been extensively studied. Therefore, the significance of this work is low. 

2) The authors' conclusion is that PPT's anti-depressive effect works by modulating the PI3K-AKT pathway and altering TTR, KL and FOS genes. However, the western blotting result in Figure 7 can not support this conclusion. 

3) Concerning the writing of the article, the abstract is way too long and need to be concise. The graphical abstract is too compact and needs to be simplified. 

Author Response

Comments 1: There have been many similar and related studies in this direction. However, the authors failed to mention them in the introduction part. 

A very recent study J. Agric. Food Chem. 2024, 72, 10376 investigates the 20(S)-Protopanaxadiol [PPD, very similar to PPT] and its antidepressive effect. 

The authors also failed to mention PPD, which has been extensively studied. Therefore, the significance of this work is low. 

Response 1: Thank you very much for your valuable comments. Although there have been many similar studies on PPD and PPT in this regard, our innovation lies in the target prediction and joint analysis of the antidepressant effects of 20(S)-Protopanaxatriol (PPT) through network pharmacology and transcriptomics. In response to your suggestions, we have researched and supplemented additional literature to support the paper, adding them as references [10] and [11] in the revised version; see lines 9 and 10 of the second paragraph of the introduction. The second paragraph of the introduction has been modified, and the modified part is highlighted in yellow, while other relevant references are also cited.

Comments 2: The authors' conclusion is that PPT's anti-depressive effect works by modulating the PI3K-AKT pathway and altering TTR, KL and FOS genes. However, the western blotting result in Figure 7 can not support this conclusion.

Response 2: We sincerely appreciate your valuable comments. In order to verify the relevant pathways and targets of PPT in the treatment of depression, we not only adopted the western blotting method, The PI3K-AKT signaling pathway, TTR, KL and FOS were systematically verified from the molecular, protein and gene levels by molecular docking (Fig. 6) and qRT-PCR (Fig. 7G-K). The molecular docking results showed that PPT had a strong affinity with the above targets, and the qPT-PCR results showed the same trend as WB results.

Comments 3: Concerning the writing of the article, the abstract is way too long and need to be concise. The graphical abstract is too compact and needs to be simplified. 

Response 3: Thank you very much for your suggestion. We have revised and deleted the abstract, and the modified and deleted part is highlighted in yellow. And the graphical abstract has been redesigned to make it look easy to understand and entertaining

Reviewer 2 Report

Comments and Suggestions for Authors

Comments:

   The manuscript describes " Network pharmacology and transcriptomics to explore the pharmacological mechanisms of 20(S)-protopanaxatriolin the treatment of depression”. Depression is one of the most common psychological disorders today. This study used a comprehensive approach combining network pharmacology and transcriptomics to explore the potential mechanism of 20(S)-protopanaxatriol (PPT) in treating depression. Use the database to predict PPT targets, and analyze the relevant targets and pathways of PPT in treating depression by integrating network pharmacology and transcriptomic analysis results. The results demonstrated that PPT significantly inhibited depression and identified 87 potential targets and 350 differentially expressed genes (DEGs) through network pharmacology and transcriptomics. It shows that PPT may treat depression by inhibiting the expression of FOS, enhancing the expression of TTR and KL, and regulating the PI3K-AKT signaling pathway, but several points need clarification.

Comment:

1. Abstract needs to be shortened

2. In Figure 1A, the mouse body weight change curve should be analyzed statistically.

3. Although this article has an abbreviated list, full names such as PI3K, AKT, TTR, KL, and FOS must be written for the first time.

4. WB quality needs to be improved.

Comments on the Quality of English Language

Minor editing of English language required

Author Response

Comments 1: Abstract needs to be shortened.

Response 1: Thank you very much for your advice. We have revised and deleted the abstract part and highlighted it in yellow in the manuscript.

Comments 2: In Figure 1A, the mouse body weight change curve should be analyzed statistically.

Response 2: Thank you for pointing this out. We have added statistical analysis to the body weight trend chart of mice in Fig. 1A and made corresponding modifications to the study of body weight change of mice in Result 3.1. The modified part is highlighted in yellow.

Comments 3: Although this article has an abbreviated list, full names such as PI3K, AKT, TTR, KL, and FOS must be written for the first time.

Response 3: Thank you for your professional comments. We overlooked this important detail and have now revised it in the manuscript, supplementing the nouns that first appear with full names and abbreviations, highlighted in yellow. Unfortunately, we have found that FOS is a nuclear phosphoprotein encoded by mature mRNA produced by c-fos gene transcription by consulting a large number of domestic and foreign literature; its full name is FOS, so there is no modification to FOS.

Comments 4: WB quality needs to be improved.

Response 4: Thank you very much for your advice. We carefully considered your comments and re-conducted the WB experiment to achieve higher-quality WB results. The modified WB is shown in Fig. 7A.

Comments 5: Minor editing of English language required.

Response 5: Thank you very much for your advice. We did our best to polish the language in the manuscript and made some changes to the manuscript. These changes will not affect the content and framework of the article. We sincerely thank you for your enthusiastic work and hope that the revised manuscript will be accepted by you.

Round 2

Reviewer 1 Report

Comments and Suggestions for Authors

The author have addressed and replied to my comments well. 

Reviewer 2 Report

Comments and Suggestions for Authors

Accepted

Comments on the Quality of English Language

Minor editing of English language required